# An Insider's Church for Outsiders: The Johannine "Come and See" Passages and Christian Engagement with the World

Michael T. McDowell 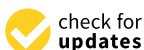

Duke Divinity School, Duke University, Durham, NC 27708, USA; michael.mcdowell@duke.edu

**Abstract:** The Gospel of John has a reputation among some New Testament scholars as a factional text designed to reinforce the Johannine Community's unity amid persecution and excommunication. Recent work, however, has proposed that John is in fact deeply ethical, with an outward-facing mission. This essay builds off this work to propose that John has a definitive missional praxis that he hopes his community will embody as it engages with the world. Examining specifically the "come and see" passages of John 1:39, 1:46, 4:29, and 11:34, this article suggests that John's method is dialectical: he simultaneously wants those in the church to remain in the church and resist assimilation with "the world", but he also wants those in the church to go into the world to understand it, empathize with it, and even befriend it, all for the sake of discipleship.

**Keywords:** John; mission; ethics; culture; Christianity; ecclesiology; missiology; ministry

## 1. Introduction

The last several years in the United States have underscored the need for a responsible hermeneutic of cultural engagement. Huge swaths of the church here have demonstrated that when Christians believe that the state is the rightful domain of the church, they will do whatever it takes to stifle dissent and wrestle the state into submission.[1] Recent revelations about the background of the events of January 6, for instance, suggest that Christian zeal played an outsized role in the turnout and motivation of the mob and its organizers.[2] However, while the events of the last several years are particularly shocking, the challenge of responsible Christian cultural engagement is not new.

The church has long struggled to define its relationship with the world. From the Patristics to *City of God* and from Niebuhr (1975) to post-colonial theology, many different visions of state–church interaction have arisen at different times. In addition, many groups still offer their own position as correct, with various kinds of accommodationists,[3] transformationalists,[4] two-kingdom acolytes,[5] and counter culturalists[6] all insisting that their way is the most Biblical.

I propose that the Gospel of John might offer some clarity to those wrestling with this question. This may come as a surprise to some readers; while John provides some rather clear missiological statements in 10:11–18, 13:12–17, 34–35, 15:12–17, and 17:6–19, he does not offer the kind of famous or obvious ethical statements of Matthew 5–7, Matthew 28, Acts 2, Romans 12–13, Ephesians 6, or James 3. As a result, and influenced by Martyn's (1968) reading of the community and its mission, some scholars have largely left John out of missiological and ethical conversations (Skinner 2017, 2020), with Matera (1996, p. 92) calling John a "major challenge" for those interested in ethics and the New Testament. Richard Hays, too, in *Moral Vision of the New Testament*, naturally privileges Paul and the Sermon on the Mount at the expense of much Johannine material (Hays 2004).[7] Matera and Hays are not anomalous; many scholars, often using Martyn's (1968) thesis that John is primarily a descriptive project, have explicitly asserted that John is an a-ethical book, with an ethic, if it has one, centered on conversion and "getting to heaven" (Sanders 1975; Meeks 1996).

Yet, others were working to show that John is, in fact, an ethical gospel. Okure (1988), Nissen (1999), and Smith (2002)—to name just a few—worked on Johannine ethics when few others were. Their work became the foundation of a newfound fascination with Johannine ethics, typified by the edited volumes of Van der Watt and Zimmermann (2012) and Brown and Skinner (2017). Three other works from Byers (2017), Gorman (2018), and Shin (2019) have more recently argued that John is a deeply ethical and missional gospel with a concrete program by which he hopes his community—and others like it—lives.

In this essay, I attempt to build off this work to suggest that John wants his community[8] to live in a dialectical relationship with the church and "the world".[9] On one end of this dialectic, the church is called to remain the church, distinct and separate from the world. However, on the other end, Jesus demonstrates that the church ought to embrace, love, and enter its surrounding communities as it seeks to make disciples, and it must do both at the same time. I will argue this thesis first by briefly reviewing the literature on Johannine ethics and mission to justify seeing John as an ethical and missional gospel. I will then focus specifically on the "come and see" passages of 1:39, 1:46, 4:29, and 11:34 to provide exegetical grounding for this dialectical methodology. Throughout this article, I keep an eye toward churches attempting to responsibly engage with their communities today, in the 21st century.

I examine these passages instead of more obvious candidates such as 10:16, 17:6–19, or 20:21–23 for a few reasons. First, the trope of "come and see" in John is understudied, especially missiologically, with one book examining its implications for discipleship (Brown 2022) and one essay looking at its implications regarding love (Do 2017). Second, while the didactic passages in John *tell* what the disciples should do, the "come and see" passages, I posit, *show* how they are to do it. As Van der Watt (2006, p. 151) explained, "Behavior is based on the interpretation of 'rules' (commandments) and express what the underlying ethical informants of these commandments are i.e., the how and why of the evident behavior". These passages, I argue, show the how and why the church is to issue—and receive—invitations to and from the world.

## 2. A New Ethical Direction

J. Louis Martyn claimed in his commentary on John that John's Gospel is so different from the synoptics because the Johannine community was so different from the synoptic communities (Martyn 1968). Whereas the synoptics likely intended for a broad audience to read them, Martyn posits, John functioned as a tool to reify a persecuted minority community. In this two-level drama, the community saw itself in the stories of the gospel, and the exegete, if they are careful, can, thus, make out the general shape and composition of the community.[10] While Bultmann (1971) and Brown (1966) each dabbled in speculation about the Johannine community, Martyn was the first to theorize about it in such specific and explicit terms. Their theory about the gospel has, thus, merited an important place in Johannine scholarship ever since. One struggles to find any serious commentary that omits careful treatment of this work.

One implication of Martyn's work concerns both ethics and missiology. If John is mostly a descriptive work, then it leaves little room for prescription. Indeed, many scholars have argued that John does not have a coherent ethic. Meier (2001, pp. 47–48) claimed that "John's Gospel is practically amoral". Sanders (1975, pp. 99–100) critiqued John saying, "Johannine Christianity is interested only in whether he [a dying person in need of help] believes". In addition, Meeks (1996, p. 318) thought "The only rule [of Jesus in John] is 'love one another,' and that rule is both vague in its application and narrowly circumscribed".

These scholars—roughly synonymous with *Gemeindeschrift*—see John's mission as primarily insular. The gospel does not exist to send people out to other communities, to save the world. Why would it? Some passages in John (3:1–18, 6:65), indicate that God will take care of the world, not the community. Instead, John was written to reify the community and encourage those within. A small, embattled community penned a gospel with striking dualism: the church against the world. The community is light; everyone else is darkness.[11]

The community has life; everyone else has death (Koester 2008). The community will never thirst, but everyone else will die parched for living water (Koester 2003). Mission exists only insofar as it keeps the community together, and, of the commands to love one another, the church bears no responsibility to love the world (Segovia 1982).

All the while, many others were laying the groundwork for a turning point in which John would properly be recognized for its ethical contributions. Okure (1988) wrote about missiology, concluding through a study of John 4 that John both is an outward-facing, missiological gospel and that it has a methodology for its mission. Nissen (1999), meanwhile, read Johannine ethics through the lens of community. He accepted Martyn's two-level premise, but, nonetheless, believed that John's vision for the community included behavior towards outsiders. Reinhartz, while not rejecting entirely the hypothesis, has long written skeptically about Martyn's two-level drama (Reinhartz 2003), with others since drawing upon Reinhartz's work to argue that John was written for many different kinds of communities (Bernier 2013). In addition, Van der Watt has worked to highlight John's ethical aspects for a long time, claiming, especially, that John's ethics are best expressed through the way that rules are "lived out" by the community (Van der Watt 2006). Others have since provided greater insight into Johannine ethics by reading them alongside other aspects of the gospel, such as time (Rahmsdorf 2019), rhetoric (Trozzo 2017), and divine action (Weyer-Menkhoff 2014).

Van der Watt and Zimmermann released an edited volume on Johannine ethics in 2012 that shined more light on the implicit ethical system of the gospel. Karakolis (2012), for instance, argued that the Johannine signs are ethical as well as Christological. Löhr (2012) similarly argued that the "works" described in the gospel have ethical implications. In addition, continuing with Brown and Skinner's (2017) edited volume on the subject, more scholars of John began to see the gospel not only as ethical, but as ethical for outsiders, too, as Gorman's (2017), Myers' (2017), and Moloney's (2017) essays discussed. And Van der Watt's (2019) new book, *A Grammar of the Ethics of John*, a systematic review and treatment of the ethics of the gospel, affirms that John does have an ethic that includes mission.

This group—*Missionschrift*—sees John and his community as evangelistic and outward-looking (though still community-focused). In this view, John intended that his gospel be spread around to all sorts of people in all sorts of other communities. Proponents of this view have often pointed to passages such as John 21, the epilogue, where Peter's catch of fish seems to symbolize the expansion of the Christian faith to those outside of the initial community (Koester 2003; Bauckham 2007). Brown (1966) reads the passage in which the Greek gentiles visit Jesus in John 12 as evidence that John does indeed intend for his people to go out to others. In addition, the mutual "sent-ness" of Jesus and the disciples in 4:38, 17:18, 17:23, and 20:21 indicates to some a missional disposition toward outsiders (Holmes 2006; Keener 2009). Under this view, then, the gospel is more evangelistic than insular.

Gorman, Byers, and Shin, in their recent books, build off this movement to propose a reading of John in which the gospel, in various theological areas, does indeed have an ethical vision for the life of the community that affects its interactions with the world.

Shin is the most general of these three scholars. He argues in his published dissertation, *Ethics in the Gospel of John* (2019), that John has a different ethical system than that of the synoptics. The synoptics communicate ethics through explicit teachings, as in the Sermon on the Mount. John, however, teaches ethics primarily through discipleship, moral progress, and imitation of stories. Many of the most famous stories in the gospel, Shin shows, have clear ethical imperatives for the reader. Over time, the reader, like the disciple, becomes more like Jesus by imitating him.[12]

Byers (2017) takes a slightly different approach, arguing that the practical aspects of John are functions of a kind of prototheosis, which he defines as "believers [being] integrated into the divine interrelation of Father and Son, generated by God himself 'from above,' and enabled to share in activities and authority readily classified as divine" (Byers 2017, p. 169). In other words, the believer becomes a child of God and then becomes

more like God. Moral formation is, thus, a part of theosis because God is good, and so the one who becomes more like God becomes better themselves, too.

While Byers takes the idea of theosis in the Gospel of John and applies it to the question of ecclesiology, Gorman (2018) applies it to the realm of mission and missiology. He specifically moves the conversation further by demonstrating that Byers' understanding of Johannine theosis can shed light on the theme of mission in John (Gorman 2018). By combining Green's (2017) "missional hermeneutic" with Byers' (2017) theotic approach, Gorman claims that John is not only missional, but also has an underlying theological vision for mission. For John, mission happens theotically, according to and imitative of the Trinity. Within the life of the Trinity, Christ both abides with the Father and the Spirit and goes into the world. By entering the world, Christ, thus, allows those in the church to do the same thing: abide and go. Followers of Christ participate in theosis by abiding in Christ and going as they do so. They sit in his love and under his instruction, and they go out and participate in Christ's work in the world.

However, thinking about receiving John in the 21st century, in a pluralist society, we might ask what it means to "go". If "abide and go" summarizes John's view of mission, what does the "go" mean and look like? The scholarship surveyed above at least allows for the possibility that John might not only have a theology of mission that applies to engagement with the world but also a method. Additionally, it is possible that this method is communicated implicitly, through stories such as the "come and see" invitations in John.

### 3. Come and See: Four Moments, Three Lessons

*3.1. Introduction to Come and See*

Few scholars have documented the parallelism of the "come and see" passages in John. Many, such as Bultmann (1971), Dodd (1998), Carson (1991), Morris (1971), and Myers (2019), make no mention of any connection between the come and see verses in their commentaries. Others acknowledge the possibility of a connection but, nevertheless, insist that no such connection exists. Brown, for instance, commenting on 11:34, admits that Jesus used the same words in 1:39, but brushes off any reading that sees a relationship between the "come and see" verses as tendentious, saying that such a reading "goes beyond the evangelist's intent" (Brown 1966, p. 426). In addition, Keener (2003) claims that the "come and see" in 11:34 "probably bears no other relation to 1:39, 46, and 4:29" (Keener 2003, p. 846).[13]

Some do see a connection. Lightfoot (1956) sees a robust thematic relationship between the three instances. He notes that in chapter 1 people come to Jesus, but in 11 Jesus goes to the people. If people move toward the light in chapter 1, Jesus moves toward the darkness in chapter 11. Culpepper (1987), too, defends reading the come and see passages together.[14] More recently, Peter Judge, summarizing the way that many scholars have taken individual occurrences of "come and see", argues that these passages are primarily Christological, even Trinitarian, invitations (Judge 2014; cf. Collins 1991).

Yet, some scholars have begun to see these invitations in John as both connected and ethically significant. Toan Do (2017), for instance, says that reading these passages as Christological invitations does not explain their repetition in various situations in which Jesus' divinity is not at all obvious, as in 1:46. Instead, the "come and see" passages are invitations to *love* Jesus. In addition, Sherri Brown (2022) agrees, showing that these verses are practical calls to discipleship that align with other passages about discipleship in the gospel. For each of these scholars, the connections between the come and see passages are intentional and intentionally ethical.

Along similar lines as Do and Brown, the reasons for seeing an intentional relationship between 1:39, 1:46, 4:29, and 11:34 are thematic, exegetical, and linguistic. Thematically, John is notable for the way he repeats and redefines different tropes and symbols throughout his book (Dodd 1998; Van Belle 2009). The duality of light and dark, for instance, first appears in the prologue and then, subsequently, throughout the gospel to provide theological context and emphasis at critical junctures (Koester 2006). Water, too, makes

several startling appearances in the book (Jones 1997), before coming to a triumphant conclusion at the cross (Koester 2003). Others have identified many more themes such as bread (Myers 2019), humiliation and kenosis (Von Balthasar 1990), royalty (Keener 2003), and glory (Brown 1966), to name just a few.

Exegetically, the "come and see" passages match the general pattern of other Johannine themes. The invitation first appears at the beginning of the book, twice in chapter 1, before disappearing for a bit. The phrase is then echoed in chapter 4, reminding the reader of the theme, before appearing fully in chapter 11 at a critical juncture—the raising of Lazarus—providing greater theological context on the significance of the sign. Furthermore, the progression of the "come and see" passages—Jesus calls his disciples, his disciples call other disciples, and Jesus draws near to the suffering of others—displays conspicuous resonance with the prologue, where Jesus "comes to his own", and "gives to those who receive him the right to become children of God", and "the Word became flesh and dwelt among us".[15] He drew others to himself, but he also went out to others.

Linguistically, John tends to recycle language to signal to the reader that a plot point or theological comment made earlier was once again relevant (McKay 1985; Van Belle 2009). The phrase "come and see" occurs several times in the gospel. The first use of the phrase occurs in 1:39, where Jesus invites his new followers: ἔρχεσθε καὶ ὄψεσθε. Here, Jesus uses the present imperative deponent form of ἔρχομαι and the future form of ὁράω, both in second-person plural. A literal rendering would, thus, read as "come and you will see", perhaps inviting the reader also to travel with Jesus over the course of the book (Morris 1971). In the second instance, in 1:46, Philip answers Nathaniel's glib dismissal with the words of Jesus, ἔρχου καὶ ἴδε. Here, Philip uses the imperative for each word, the present again for ἔρχου, and the aorist for ἴδε. However, while the forms of the words are different, the words themselves remain the same: ἔρχομαι and ὁράω. In 4:24, the Samaritan woman announces her encounter at the well with an invitation: δεῦτε ἴδετε. While the wording here is different than in the other instances, the meaning and purpose of the invitation so closely align with Philip's invitation of Nathaniel in 1:46 that 4:29 probably belongs in conversation with the other three.[16] Moreover, in 11:34–35, Mary uses the exact same words in the exact same form as Philip when speaking to Jesus about Lazarus' tomb: ἔρχου καὶ ἴδε.

| Reference | Quote | Actors | Direction of Invitation |
|---|---|---|---|
| 1:39 | ἔρχεσθε καὶ ὄψεσθε | Jesus, disciples of John | Jesus to outsiders |
| 1:46 | ἔρχου καὶ ἴδε | Philip, Nathaniel | Disciple to outsider |
| 4:29 | δεῦτε ἴδετε | Samaritan woman, crowd | Convert to outsiders |
| 11:34 | ἔρχου καὶ ἴδε | Mary, Jesus | Friend to Jesus |

Given the repetition of the "come and see" passages and their similarity to other well-documented themes in John, it seems possible that John wants the reader to somehow see these stories in the same category. But to what end? If Shin is correct, and John often communicated ethical instruction through storytelling and a call to imitation, and if Byers and Gorman are correct, and John sometimes has *specific* missiological instructions for his community, then I propose that it is possible that John hopes his community reads these passages not only as mere invitations but also as models for how to give and receive invitations themselves.

### 3.2. Come and See 1:39: Inviting the World

By 1:39, John has twice proclaimed Jesus "the Lamb of God", who would "take away the sins of the world" (1:29, 36). Two of John's disciples, hearing the messianic undertones of this declaration (Brown 1966), left John and followed Jesus. When they asked him where he was staying, he replied, "Come and you will see". Scholars have dedicated plenty of space to the intricacies of the relationship between a rabbi and his students (cf. Wenthe 2006), and, here, the students have clearly received exactly what they wanted (Morris 1971). They asked an apparently important figure where he was staying, hoping

that he would invite them to come along, which he does, even allowing them to stay with him. He extends an invitation to curious outsiders.

This is one side of the dialectic, and the first lesson that "come and see" should teach Christians about reading John in the 21st century: the church must possess a welcoming and open posture to the world. If John sees Jesus as the paradigmatic moral figure that the church is to imitate (Shin 2019), then John may be calling the church to have open doors and a welcoming posture to the world, no matter how hostile. While some would disagree (Le Grys 1998; cf. Gorman 2018), this is the posture of Jesus throughout John: anyone is welcome to come and see. Anyone is welcome to join. Anyone is welcome to investigate Jesus and his work (Gorman 2018). Even in the case of the οἱ Ἰουδαῖοι, Jesus displays an open—if harsh—posture, engaging in long, drawn out, and forceful conversations with them. In addition, when one of their "rulers" comes to him "by night",[17] he participates in an extended dialogue, explaining the intricacies of his movement and kingdom, a dialogue that bothers this ruler and draws him to the light gradually over the course of the book.[18]

Unfortunately, some churches neglect to adopt Jesus' posture by displaying a stifling suspicion of the world. This suspicion has sometimes manifested in an unmasked contempt for the church's surrounding communities and sometimes in a closed-door approach to the world. The world has nothing to say to the church; the church has nothing to learn from the world. These kinds of churches then manifest a kind of spiritual inbreeding where the community becomes so ingrown, so focused on itself, so angrily insular, that it forgets that there is anyone else around it. The same fate awaits these kinds of churches: they die.[19]

Instead, a community that takes seriously Jesus' actions in the "come and see" passages maintains a posture of openness to those that would enter. This is *not* a posture of assimilation—the church is still to be the church, as I discuss below—but it *is* a posture of curiosity and strategy about drawing in people from the world. This is the kind of posture that has led to numerous missional innovations throughout the history of the church. It is the kind of posture that led Justin Martyr to develop a *logos* Christology that would help Romans better understand Christianity (Hillar 2012). It is the kind of posture that motivated Luther to translate the Bible into German and leverage the most cutting-edge technology of the time (the printing press) to spread his hymns and Bible around Europe (Holborn 1942). It motivated Adoniram and Ann Judson to fully understand, learn, and adopt the Burmese language and customs as they preached and taught, even if only a few would ever convert (Anderson 1987). This open posture—this love, really—does not insist on its own way but is comfortable with and welcoming of the surrounding world.

### 3.3. Come and See 1:46, 4:29: The Church as the Church

However, to what end does the church welcome people? The church is more than an open door—that door must lead somewhere, and John shows where it might lead. In 1:46, Philip tries to recruit Nathaniel, using similar messianic language as had brought the first disciples to Jesus: "we have found he about whom Moses wrote in the law and the prophets: Jesus of Nazareth, from Galilee". But Nathaniel remains unimpressed, retorting, "Is it possible for anything good to come from Nazareth?" As others have shown, there may have been an understanding in first century Palestine that the Messiah—or any prophet, for that matter—could not come from Galilee (Brown 1966). The Pharisees make the same point later in the book, in 7:52. However, rather than argue the intricacies of messianic theory with Nathaniel, Philip simply says the same words as Jesus to Philip: come and see. Nathaniel does, and just one dialogue later, he has fallen to his knees and declared Jesus "the Son of God, the King of Israel".

Similarly, in 4:29, the Samaritan woman understands that Jesus is more than a Jew (4:9) and more than a prophet (4:19). Her conversation with him has so radically changed her outlook that she leaves her water bucket behind, and, where she had earlier hidden from the other inhabitants of her town by drawing water at noon (Keener 2003), she now rushes into town with an invitation and question: "come, see a man who told me everything I ever did. Could he be the Messiah?" The townspeople heed her invitation, and, when they meet

Jesus, they remain with him, like the disciples in 1:39, and they subsequently believe his testimony.

What has happened here? To use a modern term: multiplication has occurred.[20] Jesus invited in several disciples, and those disciples learned to act like Jesus, inviting others in like he did. As they spent time with Jesus, his disciples began to do the things that he did and say the same things that he said. They changed. They became like him.

This is the other side of the dialectic and the second lesson for reading John as a 21st century Christian. The open doors of the church do not lead back to the self: they lead to Jesus. No matter how open the church is to its surrounding communities, no matter the creativity it employs to open its doors and make its space a welcoming place for outsiders, it, nonetheless, must maintain a distinctive character from the rest of the world because its aims are ultimately different from those of the world: to lead, train, and teach people to know, love, and become like Christ.

Yet, even as many Christians and churches struggle with presenting a welcoming stance to the world, many also struggle with maintaining a distinctiveness from the world that leads to true transformation of its people. In the United States, one way this frequently manifests itself is in the realm of politics. Many churches on both sides of the political spectrum struggle with the fact that a distinctive witness to the world and a dedication to representing Christ to the world means that one will not and cannot fit into any one political box, especially in a two-party system.[21] Christ did and said things that confound and irritate both conservatives and liberals, and Christians on both sides of the political aisle are tempted to sanitize Jesus for their side.

Again, the Christian must resist falling too far on either end of the spectrum. Though the ultimate aims of the Christian community are at odds with the world, this is not to say that the church must always take an oppositional stance. Indeed, while there will often be areas of significance different, there also will often be areas of similarity and compatibility with the world. Even as the church maintains distance in ultimate mission from the world, it can and should use those moments of convergence to bless the world and further its mission.

The 20th century missionary, pastor, and theologian Lesslie Newbigin, for instance, spent a great deal of time writing about the need for the church to remain distinctive, especially within the realm of politics. In his time, at the height of the Cold War, the great geopolitical debate concerned communism and capitalism. However, Newbigin, at different points, insisted that Christians could not uncritically accept either system as entirely right or entirely wrong. He excoriated Marxism at times, arguing that Marxist eschatology was fundamentally incompatible with that of Christianity (Newbigin 1986). Utopianism, he noted, always leads to massacre. At the same time, he also upbraided capitalism and its uncritical proponents, noting that the accumulation of wealth for its own sake defied historic Christian teaching on the subject (Newbigin 1989). As he wrote about politics, he instead advocated for a "third way" in which Christians worked within the political system to further the common good[22] while never forgetting the powers and principalities that undergirded and stood above those same systems (Newbigin 1989). The church continues in its mission, while resisting complete isolation from the culture.

Churches, learning from Christ and his disciples here, can learn to do the same. The church will confound and confuse the world because its hope is in something higher and different than the various things in which the world places its hope, including any one political party, system, or leader.

Another way that the church has lost its distinctiveness, especially in the West, and especially within Protestantism, is in its turn towards what sociologist Christian Smith deemed "moral therapeutic deism" (Smith and Denton 2011). Scaling up from Smith and Denton's focus on youth, others have written about the prevalence of moral therapeutic deism within the church as a whole (Cosby 2012). Many churches simply try to help their people feel better about themselves and become a better version of who they were and wanted to become (Dean 2010). Rather than offering the gospel to those who entered its

doors, the church began to give people what it thought they wanted: an encouraging message only vaguely differentiable from the same kinds of messages everywhere else.

On this side of the dialectic, the church is called to maintain its missional distinctiveness from the world as it leads people to Jesus. Churches that learn from Jesus and his disciples here will learn that their work goes well beyond Sunday mornings; although it includes that, too. Churches will faithfully preach the gospel every Sunday, doing so in a way that makes sense in the language—literal and figurative—of those who might come from outside the church, demonstrating that the gospel of Jesus Christ offers something radically different and more hopeful than anything else in the world.

However, beyond Sundays, churches that maintain their distinctiveness from the world and that seek to train up their people and help them become more like Christ will have a robust small group ministry in which enterprising leaders in a variety of fields are identified, discipled, and developed such that they can start their own small groups and do the same for others. This way, pastoral work belongs to the entire church, not just one or two people. These sorts of churches would likely confound the world in a Newbiginian way with their firm adherence to Christian doctrine combined with an unrelenting dedication to religious evangelism and justice work in the communities in which they are located. In these Christian communities, the leaders know and are known deeply by their people. They eschew the trappings of hierarchy and the prestige of wider celebrity, instead endeavoring to invest in their people. In addition, these churches will slowly and consistently gather up their own members to send out to make other churches that do the same.

In short, the church remains open to anyone who wishes to come and see, but it also invites those who come and see to come and see more deeply, until those people, too, are so trained up in Christ that they begin using his same words to others.

### 3.4. Come and See 11:34–35: Going and Seeing

As mentioned above, the "come and see" passages follow the same pattern as many Johannine themes: a theme is introduced, further defined, and later redefined. In this case, the first instance appears to concern welcoming in those from the outside and the second with teaching those on the inside to do the same things as Jesus. Now, in the third repetition, the theme finds it apogee in the culminating moment of the first half of the gospel: the raising of Lazarus.

Jesus, John notes, delays from visiting Lazarus, a puzzling move that only makes sense within the context of his growing mission (O'Day 1995). When Jesus arrives, both Mary and Martha demonstrate their grief: Martha in going out to meet him and Mary by staying back and falling to her knees. However, they each make the same accusation: "If you had been here, my brother would not have died" (11:21, 32). The two characters thus become "blurred" in a haze of grief of confusion (Myers 2019, p. 134).[23] And Jesus is not immune to these emotions. When, "deeply moved",[24] he asks where the body is, Mary tells him to "come and see" and continues weeping. Jesus does come and see, and he weeps too.

Though the crowd interprets Jesus' tears as borne out of grief because of his love for Lazarus (11:36), some scholars have asserted otherwise. Moloney (1998), for instance, thought that Jesus' tears were borne out of frustration at the inability of Mary and Martha to understand what was about to happen. Many, such as Brown (1966), Beasley-Murray (1999), and Mensah (2017) have noted the use of the Greek word ἐμβριμάομαι here and concluded that Jesus was not just frustrated: he was *angry*. Some then see his tears as the outflowing of that anger, be it directed at Mary and Martha for their unbelief (Morris 1971; Mensah 2017) or death itself (Brown 1966; Lee 2010). Lee (2010) also understands the tears as the outward manifestation of Jesus' realization that raising Lazarus would lead to his own death.

However, some also read Jesus' tears as related to grief. Brown (1966) discusses a Jesus overcome by anger *and* sorrow. Carson (1991), likewise, sees Jesus' tears as symptomatic of a person overwhelmed by grief and rage. While cautioning against "sentimentalizing Jesus' tears", O'Day writes that "these tears are thus positioned in the story as Jesus' public

acknowledgement of the pain that death causes in human life" (O'Day 1995, p. 691). Leaving aside the question of his anger, Jesus' weeping right after Mary wept, his clear love of Lazarus, and the response of the crowd seem to indicate that at least *part* of the cause of his tears was grief. This, then, is significant. Jesus did not *need* to grieve, but he did. He knew what was going to happen, but instead of standing above the fray, he accepts Mary's invitation to "come and see", and when he does, he grieves with her. As Myers (2019, p. 135) writes,

> Jesus matches Mary's physical display of devotion with one of his own. He observes her "weeping" (klaiō, cf. 20:11–15) along with the "weeping" of the Jews around her, and he is "deeply moved in spirit and stirred in himself", so much so that he himself "cried" even though he knows that he has come to "wake up" Lazarus (11:11, 33–34). The suffering that Martha, Mary, and the Jews experience is not insignificant to Jesus, or to the Father, whose will and perspective Jesus manifests. Jesus becomes a mourner alongside Mary and the Jews regardless of the end of the story and Lazarus's rising from the dead.

The "come and see" here seems to function as a double entendre: Jesus *eventually* comes and sees the body, but he *first* comes and sees something different: the pain, grief, and suffering of those close to Lazarus. He weeps at Mary's invitation, not because he does not know what would happen, but because he so deeply loved Mary, Martha, and Lazarus. John apparently sees this moment as significant enough to warrant comment, using the crowd almost as a Greek chorus[25] to offer differing interpretations of Jesus' own tears. This is not the austere, borderline-stoic Jesus that sometimes dominates in other moments in John (Attridge 2010): this is a Jesus that truly "dwells among us". In line with a theotic interpretation of John, Jesus does more than offer invitations to become like him: he becomes like those he invites (Byers 2017). Thus, Jesus weeps.

This iteration of "come and see" differs from the others for another reason, too. The opposite party in this exchange, Mary, is a friend of Jesus, not a stranger meeting Jesus for the first time as in 1:39, 1:46, and 4:29. However, given the clear structure of these other invitations (outsiders and insiders) and the repetition of the phrase here, the reader should consider whether John still wants us to see Mary *in some sense* as an outsider. Dorothy Lee (2010, p. 1164) reads the women this way, saying,

> Though a 'friend' of Jesus and therefore a disciple, Martha still does not understand the full implications of his identity. Despite some optimism, she assumes Lazarus is beyond hope and that resurrection is a future, postponed reality . . . Jesus subsequent meeting with Mary and the mourners shows again their lack of understanding.

While Mary and Martha, then, are not "outsiders" in the sense that they do not know Jesus, they *are* outsiders in that they do not yet fully *understand* Jesus.

Read this way, this moment represents a redefinition of the first "come and see", for Jesus not only welcomes others into his circle, but he actively goes out and understands others, too. If the church is called to imitate Jesus, it is also called to dwell on earth, and dwell so much so that it deeply understands, sympathizes with, and goes out to the surrounding world. The church cannot merely sit on its own hill in its own building, however wide open the community flings the doors. Rather, the church must go into the world, seeking to understand culture and seeking to articulate the stories it tells itself and the desires it has so well that those who walk into the church and hear the pastor feel as though the pastor is preaching directly to them. Those in the church are called to a project of empathy: to feel what those in the surrounding areas feel, even if they know that something higher is out there. To mourn where many mourn, even if they know that there is hope. To understand what they want, even if the church understands that these desires are too small.

However, to do these things, the church must go out. To empathize with the culture, to understand it, and to earn the right to speak to it, challenge it, and compel it (Keller 2012),

disciples of Jesus must make a concerted effort to live outside the church. Churches should encourage its people to befriend non-Christians—genuinely befriend them, though not in a manipulative or utilitarian way—and leaders of Christian community should lead the way in doing so. Churches should help their people identify a "third place"[26] in their lives, be it a gym, a coffee shop, a club for people with similar hobbies, or even a playground for their children. In these places, Christians can unobtrusively and genuinely meet and befriend people who are not Christians. In an increasingly polarized 21st century, it is all too common for Christians to find themselves in a silo with other Christians. Christian friendship is indeed important, but so is friendship with non-Christians.

Recent psychological research suggests that people's stereotypes and prejudices towards a specific group break only when they personally know someone within that group (Broockman and Kalla 2016). Many people view Christians as bigoted, prejudiced, judgmental, and regressive, sometimes rightly so. This view only changes if people begin to have friends who are Christians: who have eaten with them, played games with them, watched their kids with them, traveled with them, laughed with them, and cried with them. Only then might Christians earn the right to say something to those outside the church and only then might those outside the church hear them and listen. Only then, of course, might Christians have an open door to share that deeper hope that they have about resurrection and life and about truth and light, because only then will Christians understand the world, and only then will the world see that Christians understand them and love them. Only then can Christians begin the work of discipleship.

## 4. Discussion and Conclusions

In this admittedly interdisciplinary essay, I have proposed that John has a concrete method for engaging with one's surrounding communities modelled by the "come and see" passages in the book. I first discussed scholarship, older and more recent, on Johannine missiology, arguing that a better reading of John understands it as a fundamentally ethical, outward-facing, and evangelistic book. I especially reviewed Gorman, Beyers, and Shin, to understand how John views his gospel not just as a story or a mirror, but rather as a proposal for how readers and members of the Johannine community ought to live.

Authorial intent is often hard—if not impossible—to "prove". However, after laying the groundwork for understanding Jesus as a figure worthy of imitation in John and alleging that the "come and see" passages of 1:39, 1:46, 4:29, and 11:34 ought to be read in tandem, I proposed that John uses these passages as examples of a dialectic for engaging with and speaking to the world.

First, the Christian community must leave its doors open for others outside the church to enter. It should, especially, set up its Sunday services, so that those outside the church feel comfortable and welcome.

Second, the Christian community must remain the Christian community by focusing first and foremost on making disciples of those who enter the doors of the church. The open door leads to a journey of hearing the gospel, following Christ, taking the sacrament, and differentiating oneself from the surrounding world. In particular, the church should never become so enamored with power that it uncritically drifts down a path of political domination, shilling for one side over another, so that people outside the congregation see the church as just one more partisan entity. Instead, the church draws in and disciples others by affirming what is good in a culture and challenging what is bad.

Third, the Christian community should come and see the world as well. It does not only invite people in, but it also goes out itself. Christians should befriend non-Christians, doing the work to understand, empathize with, and engage with those outside the walls of the church. Rather than hemming in oneself in an endless series of church events, the Christian should spend the bulk of their social time with people who are not Christians. Only in this way will the Christian understand the world and those in it, such that they can effectively compel and challenge them and bring them into those open doors of the church.



While I have focused on the "come and see" invitations as modeling this dialectic, other passages in the gospel support this reading too. John 17:15–18 is especially relevant:

> I am not asking you to take them out of the world, but I ask you to protect them from the evil one. They do not belong to the world, just as I do not belong to the world. Sanctify them in truth; your word is truth. As you have sent me into the world, so I have sent them into the world.

Jesus here describes the dialectic I have discussed in this essay. He tells the Father that the disciples do not belong to the world. They are separate and different. However, at the same time, they belong in the world, at least for the time being, and the Father should not take them out of the world. They are to both go into the world, but they also are to remain distinct from the world, and they are to do both at the same time.

In a 21st century marked—especially in the West—by an increasingly caustic political and social environment, churches that both resist total withdrawal and uncritical alignment with any one side in a culture war stand to offer a compelling witness to the world. If the world sees churches as bastions of quietist holiness or as silos of political sentiment, it may not have much interest in listening to the church. Not at this moment. However, if it sees churches that earnestly work to better their communities, befriend others regardless of background, and genuinely welcome curious outsiders, then the church may earn enough trust with some to begin the process of discipleship. At the very least, this posture keeps the church from dominating others in the culture and politics without pretending that real issues have no import for the church.

This balancing act—to be the church while loving those in the world—this is the true challenge of the community of believers in John, and so it is today. Christians must seek to love—to genuinely love—the world and those in it, but they must also seek to be the church, to be different, and to embody the greater hope that they have in Jesus Christ.

**Funding:** This research received no external funding.

**Conflicts of Interest:** The author declares no conflict of interest.

## Notes

1. See this speech from Lauren Boebert, United States Representative from Colorado, in which she says that "I'm tired of this separation of church and state junk" and "the church is supposed to direct the government". https://twitter.com/patriottakes/status/1541508454740885511?s=20&t=-POZf91rejzTSnLKUrF3BQ (accessed on 5 July 2022).

2. The commission presented a video as its opening statement showing, among many other things, Christian flags flying alongside American flags within the mob. https://www.google.com/url?sa=t&rct=j&q=&esrc=s&source=web&cd=&cad=rja&uact=8&ved=2ahUKEwjk9fnY8tT4AhUDuKQKHdGsA-wQtwJ6BAgJEAI&url=https%3A%2F%2Fwww.youtube.com%2Fwatch%3Fv%3Db3_O91gyj9o&usg=AOvVaw218LUI-0KsWkYOIu2pdfY3 (accessed on 5 July 2022). (cf. PBS 2022).

3. Many so-called "seeker sensitive" churches such as Willow Creek have faced criticism that they are really accommodationist. (Cf. Pritchard 1996).

4. Typified especially within Reformed Circles by Abraham Kuyper, most succinctly in their address at the Free University Amsterdam. in which they asserted, "There is not a square inch in the whole of creation over which Christ, who is Sovereign over *all*, does not cry: 'Mine!'" (Kuyper 1880).

5. A mostly Lutheran doctrine, explicated well in Stephenson (1981).

6. Famously expressed by Mennonites, especially in John Howard Yoder's *The Politics of Jesus* (Yoder 1994).

7. Hays does not ignore John—he devotes an entire chapter to John's Gospel and letters, after all—but his Scripture index shows how much he privileges the older Synoptic material and Paul, especially Romans.

8. While I do not mean to imply a total one-to-one relationship between the Johannine community and modern churches, the latter represent contemporary communities that have received the gospel, like John's millenia ago. Churches that view John's instructions as important and/or inspired can, thus, heed the directions he gives his community.

9. The word "κοσμος" appears 78 times in the Gospel of John, most of the time referring to the institutions, groups, and events occuring outside the Johannine community. I will use its literal rendering, "world" in this essay. Cf. Braun (1965); Cassem (1972); Marrow (2002).

10. Martyn certainly believes that the exegete must be careful. One reason this book is so influential is Martyn's skill as a writer; it is a delight to read him. For just one example on these pages: "it has occurred to me that it would be a valuable practice for

the historian to rise each morning saying to himself [sic] three times slowly and with emphasis, 'I do not know'" (Martyn 1968, p. 146).

11　For more on dualism in John, see Smith (1995, pp. 42, 69, 85) and, more recently, Bauckham (2015, pp. 119–25).

12　Cf. Moreover, Bennema's recent work (Bennema 2017).

13　Although he earlier emphasizes the parallel of 4:29 with 1:46.

14　He does not touch on 11:34 but does note the connection.

15　Beasley-Murray speaks for many scholars when he says in his commentary, "For the Evangelist it would appear that the account of the prologue moves to the statement of v. 14; by virtue of its theological significance *it* forms the center of gravity of the prologue, and indeed of the Gospel itself". (italics original). Beasley-Murray, *John*, 4.

16　See Keener's enthusiastic defense of this point in his (2003) commentary.

17　Showing perhaps Nicodemus' then-current status as belonging to the darkness. Cf. Beasley-Murray (1999, pp. 358–60).

18　Again, see Beasley-Murray (1999, pp. 358–60) for a lengthy treatment on Nicodemus' development over the course of the gospel.

19　Since it is very hard to tease out differences between Evangelicals and Fundamentalists in surveys (with some sociologists alleging that the distinction is no longer meaningful), it is difficult to find any concrete data about the decline of fundamentalism alone. However, the numbers from the most conservative denominations suggest a general trend. The Southern Baptists have lost over a million members over the last three years and are now at their lowest membership number in 40 years, and 2021 marked the lowest membership in four years for the Presbyterian Church in America (PCA 2021). A recent survey found that nearly 60% of Free Will Baptist churches are either in decline or stagnant, with 80% of members having grown up in their respective church. The pattern is clear: the most conservative denominations are now following the mainline in decline. Shellnutt (2022); NAFWB Committee on Denominational Research (2020).

20　For one of the many books about church planting, discipleship, and multiplication, see Payne (2015).

21　This topic takes up about half of Keller's short book on reaching the West. Keller (2020).

22　Newbigin calls Christians "patient revolutionaries". Newbigin (1986, p. 209).

23　Elizabeth Schrader's (2017) suggests that this blurring was an intentional move by a later editor to reduce Mary's role in the Gospel.

24　For more on the Greek here, see Beasley-Murray (1999, pp. 183, 192–93).

25　For more on the parallels of crowds and Greek chorus in John, including chapter 11, see Parsenios (2010, pp. 54–64). Moreover, Brant (2004).

26　Not my own term. See Oldenburg and Brissett (1982).

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
