# Peer review of "An Insider’s Church for Outsiders: The Johannine “Come and See” Passages and Christian Engagement with the World"

_religions, doi:10.3390/rel13090865_

Round 1
Reviewer 1 Report
There are several in accuracies throughout the document. To mention some: line 36 mentions 'Keller 2019' but there is no such work in the References section; in n. 8 it is 'Alan' not 'Allen', and there should be a space between 'C.Clifton'; n. 9 does not mention the page number for the quotation; n.14 it is 'Cornelis' not 'Cornelius'. And so on.
Jonathan Bernier has most thoroughly refuted Martyn's hypothesis. J. Bernier, Aposynagōgos and the Historical Jesus in John, BINS 122 (Leiden: Brill, 2013).
I have difficulties with the author's Section 3, where s/he explains his/her understanding of the three 'come and see' passages in John 1:39, 46; 11:34. Contra the author's claim in line 522, it is not John who uses these three passages as examples of cultural engagement, but the author of this article. The author applies an allegorical reading rather than an exegetical reading of 1:39, 46; 11:34. The author briefly looks at each passage, which then becomes quickly a launching pad for an allegorical or ideological reading of cultural engagement. If the editors are happy with this approach, then that's fine but I contend that it is a reading imposed on the text rather than coming out of or driven by the text. For example, regarding 1:39, the author argues that the church can or should learn from culture, but that is not the point of 1:39. While Jesus does invite outsiders to come and have a look at him, the point is NOT that Jesus was to learn from the two outsiders. Jesus' point was not that he should learn to be open to culture; rather, he invited outsiders to learn from him. Then, regarding 1:46, while I agree with the very brief explanation in lines 353-357, the author then goes on to speak of the church's need to change (lines 358-425), but this seems more an allegorical/ideological reading than a reading that is rooted in the text. I find it difficult to accept that this is John's intention (see the author's claim in lines 76-79, 522-523). If the author claims that this is simply his/hers ideological reading of the Johannine text, it may work. And I maintain that the author's reading of 1:39 that the church should be open to culture is not the point of 1:39.
Reviewer 2 Report
Page 1 , line 27 “nee” is usually used of a female name before marriage. It is not, therefore, appropriate for Caesar
p. 2. Lines 43-44. The absence of John from missional discussions is an odd comment given its relationship to missio Dei, and the writings of Michael Gorman and Teresa Okure, to name but two….whom the author later cites (Gorman). This is symptomatic of a tendency to make fairly bald statements which need qualifying- and. That fact is then recognised by the writer’s own citation of. Contrary viewpoints. So yes, the cultic reading of the community was influential, but there were always dissenting voices. In some ways , Culpepper’s earlier comparison of the community to a school indicates that.
I would strongly suggest that the writer also needs to mention John 3:1-18. Saving the cosmos is not usually a cult activity. I realise that constraints of space preclude a detailed study, but even a whiff of a reference might help the case.
p. 2. 48. Need to differentiate UK and US understandings of sect and cult.
p. 2 55. Note also that more recently attempts to reconstruct the community are increasingly ,marginalised and the focus returns to the text.
In some ways, what has been set up in the introduction is a straw woman. Almost every point in it may be qualified by reference to other scholarship
4. 106- the “indeed likely was mostly” is a catena of qualifications- and different views need to be cited.
-pp4-5 134-137 needs perhaps to reflect more on the philosophic homoiosis theo and divinisation in the ancient world ( thus David M. Litwa). Note that Platonic patterns which were in circulation at that time have a mystical element……
“theosis proper has more of a mystical element to it than pure discipleship or moral formation.”
begs a number of questions , and risks being interpreted through an anachronistic Christian lens, or a circular argument. “Simple” might be better than pure, but even that begs the question of what constitutes “discipleship”
A definition of theosis is needed, I would suggest……
5. 143 esp. as this passages suggest a movement INTO belief, which makes less sense if directed to those who are already in…..A sectarian text might be expected to say more about not leaving, than entering…….
p. 5 170- might be worth. Describing who the Greeks are- Greek speaking Jews, or Hellenes. This will affect the scope of mission.
p. 6 223. Might the sentence finish with “John.” “Honest exegesis” sounds unnecessarily harsh, esp when the following sentence admits that the difference of approach to narrative/literary makes a difference.
225 is (tepid) warranted? It may be my own choice of tone, but, the use of adjectives like this heads towards ad hominem argumentation, and lays the writer open to the criticisms made by Margaret Michell (in New testament in Comparison, (ed Barclay and White) of C. Kavin Rowe’s One True Life. I view it as unnecessary. Please feel free to disagree.
7 233-34 too many “themes” (adv and object)
242 “come and see” in commas?
8 276 Might the difference in word have anything to do with the. character who offers the invitation. Coming and Seeing as intimated by Jesus, which points to himself, will have a different shape from that by the Woman, which will point from herself elsewhere- to Jesus.
295- the reference to Thiering is a red flag- she is discredited as a source. The same point could be made from less contentious sources, and ideally with cross-references to primary sources, even if they are complex in their handling. Consult Steve mason on Josephus for one, and his of this as source material. Again there is a problem lurking here- we do not know of any prerequisites for entry to Johannine circles.
302. church or community. Is there a need to distinguish here the community of the audience and the modern church…..? it may be my own foible, but church seems to be blending two disparate groups ( the community who received the gospel and the modern church) together….
312. A stronger opening sentence? Something like” Regrettably, the church has all too frequently failed to adopt the manners of Jesus. It has….”
Again, as a missiologist, I would point out that this is a rhetorical generalisation. There have been honourable exceptions. Matteo Ricci and John Colenso are two immediate examples. They are not alone, even in ancient (Acts 17), medieval and modern times.
320-21 I am not sure what the “same thing” refers to. I suspect just the earlier part of the paragraph but it had me looking further back.
322- church or community.
325 onwards- again the writer’s own word tend to qualify the very blunt rhetorical picture of the church from which their comments start. This section could be buttressed by a number of examples from church history, not just the writer’s experience ( fn 25). The. business of translation (Jerome, Cyril, Methodius, Krapf, Steere, Henry Martyn). Cut to the chase- Lamin Sanneh, Translating the Message (2nd. 2009)
349- prophets and Galilee- see Jonah (2 Kings 14:25); Elijah ? (1 Kings 17:1); Nahum 1:1…are the claims made by Jesus’ opponents a mark of bluff, ignorance, or. Difference from the Jewish Scriptures? Ultimately, this may be redundant, as the author notes, the claim is simply ignored, not refuted.
363- but this. claim of fundamental difference. Is too simplistic- there may be points of divergence and convergence. To stress difference here moves to place Jesus is conflict with, for example, the Temple. What may be happening, and we could argue this, is that there is significant convergence in terms of Jesus now revealed as the one who actually fulfils what the temple aspires to do. These remarks. Need to engage with the. complexity of Niebuhr and those who followed in his wake. Ironically, the author risks setting up the kind of binary opposition between. Church and culture which has been criticised earlier.
392- but there is again a problem of opposites. Some of what. Both capitalism and Marxism claim as. Ideal, or worthy resonates with what. Christian values state. The problem is that they are not co-terminus. This passage, which dwells only on Newbiggin, deserves more work on political theology to avoid the charge of being an uncritical citation of a single writer.
426 onwards - surely a thematic discussion of water should also include 7:37-39, especially as these verse read best as identifying Jesus as the source of that water……This would strengthen the example.
450 – does the introduction of the book of Signs. Raise more problems than it assists? After all, you have got this far without mentioning it…..and the. business of Johannine sources is not germane to the. narrative literary approach pursued so far.
461- are you sure that this is the only interpretation. Grief is problematic because of the introduction to Chapter 11. Moloney argues for frustration, Dorothy Lee for a proleptic revelation of what awaits Jesus
465. see also Jo-Ann Brant Dialogue and Drama (2004)
474 onwards- as the thesis and application depend on identifying the tears with grief, this case should be made in the earlier section, not simply assumed. The paragraph makes the mission of the church dependent on empathy- is this really the outcome sought….? After all, faith involves elements of epistemology, relationship and allegiance. This seems to prioritise one element. Feel free to disagree.
486 onwards- might you not stress that the effectiveness of Jesus’ ministry in Chapter 11 in apart. Stems from his prior friendship with the family group, even though this has not been clearly set out in the narrative. The paragraph finishes with comments a bout a Christian silo. Let me play Devil’s Advocate…..Are not Martha and Mary already WITHIN Jesus’ silo? Wouldn’t. the author’s point come stronger from say the healing of the blind man, who was not at first in the silo, but does become a Christian?
Final thoughts
- welcoming is fine , but it is only a first step in the business of making disciples.
Tendency to make selective and bold statements which are then contradicted by own research- e.g., Martyn on cult and descriptions of past and contemporary missional practice. The lack of historical reference to. the mission practice. Outlined should be rectified, and. Exceptions to the caricature noted. The historical missiological knowledge shown is minimal.
The writer needs to recognise differences of opinion to views stated in the paper- e.g., John 11 and Jesus’ tears. As the paraenetic material addressed to the church depends on this reading it should be rigorously made, but there is no evidence of awareness of different interpretations (see line numbered comments)
The paraenetic material risks reaching the conclusion that good welcoming is all that is needed. Is there not more to the making of disciples?
The bibliography is good, but the text is obviously dependent on some works cited (e.g. Keener) more than others.
There are some needlessly critical adjectives used in the description of some scholars’ work.
Reviewer 3 Report
A brief summary:
I really enjoyed reading this article. It was easy to follow. It fits the theme of the special issue (“reading John’s Gospel” and “contemporary reflection”). It is applicable to contemporary questions (and tensions) in missional theology. It was well-structured.
The theme of the article, that a Johannine missional mindset includes an invitation to those in culture to “come and see” but also a willingness for the church to go to those embedded within the culture with compassion, comes through consistently, from the abstract, through the body, to the conclusion. The basic thesis of this dual approach is reasonable, and seems relevant to contemporary missional concerns. I am sure many readers would enjoy and appreciate the missional insights and the cultural engagement approach of tension and balance that is presented.
General concept comments: Things that could be addressed:
The subtitle of the article is “a Johannine Model of Christian Engagement with Culture.”
First, if the purpose is to explain a “Johannine model” of missional mindset, there are other texts and concepts in John worthy of consideration that more directly relate to “mission” in John (than John 11:34). Some things to consider:
Lines 144-145: “it is hard to explain away 20:30-31”
Yes, but to add to that, John 20:21-23 earlier in that context of chapter 20 takes a key “just as” Johannine construction, used with other concepts (“as the Father has loved me, so have I loved you”), and applies it to a mission concept: “As the Father has sent me, I also send you.” And John 17:18 directly teaches, “As You sent me into the world, I also have sent them into the world,” that the world may also know that the Father sent Christ, “and have loved them as you have loved me” (John 17:21, 23). Also John 10:16 would seem to pertinent (especially when combined with John 21:15-18), but I do not believe it is ever cited in the essay: “And other sheep I have which are not of this fold; them also I must bring, and they will hear My voice; and there will be one flock and one shepherd.” These are more explicitly missional materials than John 11:34?
Lines 215-220: To be fair to the contention of Keener (and similar contentions), John 4:29 is similar to John 1:39, 46 in that the narrative contexts are both about being invited to experience him firsthand (whether Andrew by Jesus himself, or Philip being beckoned by Nathaniel, or the Samaritan village residents by the Samaritan woman). Those individuals are invited to come to see Jesus (a very missional action). While in John 11:34, it is Mary and her onlooking companion “Jews” who tell Jesus, “Come and see.” Yes, John 11:34 has the exact parallel of vocabulary of erxou (although John 4:29 is similar to the LXX of Ps 66[65]:5), but conceptually John 4:29 is similar in movement and purpose. Being invited to come and experience Jesus firsthand. It is unclear on a prima facie level how John 11:34 is as “missional”—The use of the text (and reversing the “Come and see” of the believers having open and compassion interaction with members of the culture) works well as a missional principle, but it’s not as clear that this is the authorial intent in context. So, it works well as an example of reading/reception (and thus fitting this specific, themed issue of Religions). but not as convincing as an example of authorial intention. If this is true, the abstract and the thread of wording through the article should be more along the lines of drawing from the “come and see” phrase in John 11:34 as well as John 1:39, 46 (which the reader is able to do), rather than the claim that this is what John 11:34 actually meant to teach by authorial attention (a goal that is more difficult to raise to a high level of confidence). In any case, the Toan Do article mentioned below, that merges the “come and see” discussion of John 4:29 with John 1:39, 46 and John 11:34, may also be relevant.
If one wanted to deal more with the clear intentions of a more didactic approach (the purposeful teaching of the Gospel of John), a clear thread would be Jesus being sent into the world / coming in the world / being the light of the world, and then sending his disciples (John 20:21-23) who are now “in the world” even though he is no longer in the world (John 17:11). As John 17:18 directly teaches, “As You sent me into the world, I also have sent them into the world,” that the world may also know that the Father sent Christ, “and have loved them as you have loved me” (John 17:21, 23). Even though they are “not of the world,” Jesus does not pray that they would be taken out of the world (John 17:14-15), leading to a missional tension of being in the world but not of the world. Is not this the very balancing act in the conclusion of the article, lines 548-550: “This balancing act—to be the church while loving the culture—this is the true challenge of the community of believers in John, and so it is with us.” Thus would the sentiment of being in the world but not of the world be relevant?
Second, some thoughts circling around placing the modern anthropological-sociological construct of “culture” upon the Gospel of John. Some things to consider:
In the Abstract:
Lines 11-12: “but he also wants the church to go into the surrounding culture, to understand it, empathize with it, and even befriend it.” What is meant by “culture” throughout the article? Pop culture or high culture? Mainstream culture or minority subcultures? Sociologically, a culture can be construed as the developed product of human creativity viewed as worthy of transmission through socialization (including expected behaviors and norms). It is therefore inherently axiological as an outworking of an ordering of values.
So to invite in and befriend “culture” to what end? To be open and welcoming, yes, but ultimately to seek for people within the culture to be transformed in their values or to remain embedded within them? Moreover, in preciseness, to befriend the culture, or to befriend those within the culture? It often seems that “culture” in the article is simply a way of referring to people within the surrounding culture, which does fit the personal interaction themes of the Gospel of John. In John 1, the “come and see” is an invitation to people in the environs to come and experience Jesus firsthand. Thus, perhaps, going back to the Abstract: “but he also wants the church to go into the surrounding culture, to understand, empathize with, and even befriend those within it.”
Lines 534-535: “And third, the Christian community should come and see the culture as well. It does not only invite people in, but it also goes out itself. Christians should befriend non-Christians, doing the work to understand, empathize with, and engage with those outside the walls of the church.” Here, the befriending, understanding, empathizing, and engaging is the people (after the passing reference to “the culture as well”)—a clearer focus of the verbs than the modern anthropological-sociological construction of “culture.”
The Johannine writings themselves do not directly discuss the concept of “culture.” They do interact with a somewhat related (although not synonymous) concept of kosmos. In Johannine theology, the “world” can be the people (“for God so loved the world”), but it can also be an axiological system of values and desires (“love not the world, neither the things that are in the world”). It would seem perhaps that the Johannine ideal would be to avoid imbibing the "world's" valuation system, even while being sent in compassion to love the people embedded within it? How does this Johannine concept of kosmos overlap (or not) with the contemporary concept of “culture”?
Lines 12-15: “I argue that the ‘come and see’ passages of John 1:39, 1:46, and 11:34 typify this instruction as the reader, seeking to imitate Jesus, learns that they ought to invite in the culture, maintain distance from the culture, and go to the culture.” Is John 11:34 is really about “the culture” in context (where Jesus is being asked to “come and see” the place where Lazarus had been laid)?
In sum, “come and see” as a window to cultural engagement within John is a good example of “reading John’s Gospel” and of “contemporary reception,” but to say the Gospel author intended the reception insights (concerning a modern anthropological-sociological construct of “culture”) regarding cultural engagement (esp. in 11:34) is a harder task. Perhaps be consistent with the “reading of” and “contemporary reception of” John focus and language and tenor? Acknowledging that one can draw from the Gospel for one's purposes of our modern interest regarding engagement with "culture," but the author of the Fourth Gospel may not have been thinking with the frameworks of modern anthropological-sociological constructs.
Third, the Bibliography and relevant resources: Some things to do consider:
Lines 74-76: “But first, I will discuss the new ethical direction of Byers, Gorman, and Shin to justify seeing John as 75 possessing a concrete missiological method.”
And Lines 516-519: “I suggested that John is also a deeply ethical book, using Gorman, Beyers, and Shin as starting points to understand how John views his Gospel not just as a story or a mirror, but rather as a proposal for how its readers and members of the Johannine community ought to live.”
The article highlights the works of Byers, Gorman, and Shin. A key turning point in focusing upon Johannine ethics as a field worthy of study came with the 2012 work of van der Watt and Zimmermann, leading to many fruitful offshoots, and followed by the likes of Christopher Skinner and Sherri Brown and Cornelis Bennema. Bennema’s major monograph is cited after the sentence ending in line 130. And footnote 18 cites Sherri Brown’s recent monograph. There are locations within the article where a familiarity with other key works would be warranted/ For example, lines 124-126 mention the imitation and narrative themes of Johannine ethics: “In John, however, the ethical model comes primarily through discipleship, moral progress, and imitation of stories, the latter especially being well-known in the Greco-Roman world.” In any case, an ATLA database search for Johannine ethics materials cross-referenced with the scholar’s names I reference above (and others, like Dirk van der Merwe) leads to such influential books as those below (not even touching the many articles):
van der Watt, Jan, and Ruben Zimmermann, eds. 2012. Rethinking the Ethics of John: “Implicit Ethics” in the Johannine Writings. Tübingen: Mohr Siebeck.
van der Watt, Jan G. 2019. A Grammar of the Ethics of John: Reading John from an Ethical Perspective. Tübingen: Mohr Siebeck.
Brown, Sherri and Christopher W. Skinner (eds.). Johannine Ethics: The Moral World of the Gospel and Epistles of John. Minneapolis: Fortress Press.
These above would seem key. Others to perhaps consider (though perhaps not as critical):
Rahmsdorf, Olivia L. 2019b. Zeit und Ethik im Johannesevangelium: Theoretische, Methodische und Exegetische Annäherungen an die Gunst der Stunde.
Trozzo, Lindsey M. 2017a. Exploring Johannine Ethics: A Rhetorical Approach to Moral Efficacy in the Fourth Gospel Narrative. Tübingen: Mohr Siebeck.
Weyer-Menkhoff, Karl. 2014. Die Ethik des Johannesevangeliums im Sprachlichen Feld des Handelns. Tübingen: Mohr Siebeck.
In particular, an essay in the Brown & Skinner Johannine Ethics collected volume mentioned above is especially relevant to this article submission’s presentation:
Do, Toan. “The Johannine Request to ‘Come and See’ and an Ethic of Love,” in Sherri Brown and Christopher Skinner (eds.), Johannine Ethics: The Moral World of the Gospel and Epistles of John. Minneapolis: Fortress, 2017, pp. 177-196.
Another relevant resource worthy of inclusion, on the missional side (rather than the ethics side), is the following (entailing an openness to viewing John 4 as missional in orientation):
Okure, Teresa. 1988. The Johannine Approach to Mission: A Contextual Study of John 4:1-42. WUNT 2.31. Tubingen: Mohr Siebeck.
Specific comments:
Typos, etc.
Line Explanation
15 the 21st Century
See: https://www.chicagomanualofstyle.org/qanda/data/faq/topics/Capitalization/faq0009.html
the 21st century
107 and it indeed likely did face all sorts of persecution
This wording, at least to me, seems redundant. Maybe:
And it likely did face all sorts of persecution
120 He argues in his dissertation
Since the 2019 date provided is the publication of Shin’s work by Brill:
He argues in his published dissertation
162-163 it keeps the community safe and stolid.
Word choice? “stolid” usually means “showing little or no emotion or interest”
177-178 but also has underlying theological vision for mission
but also has an underlying theological vision for mission
206 any connection between the come and see verses
to be consistent with the punctuation elsewhere, as in lines 203-204 immediately above:
any connection between the “come and see” verses
257, 262, 264, etc.
Greek accenting (not just breathing marks, but accenting)?
284-285 Scholars have dedicated plenty of space of the intricacies of the relationship
One dedicates space “to” something
Scholars have dedicated plenty of space to the intricacies of the relationship
358-359 as a 21st Century Christian
See capitalization of “century” above, but also when the number is merged as an adjective, then hyphenate?
As a 21st-century Christian
375 The 20th Century missionary
The same:
The 20th-century missionary
384 Capitalism
to be consistent with the capitalization in line 378 above
capitalism
442 to those he would call His own
to be consistent with capitalization elsewhere (including he / His here)
to those he would call his own
587 H Richard Niebuhr. 1975.
List and alphabetize with the surname first:
Niebuhr, H. Richard. 1975
592 Zondervan, Cop
Unsure of what “Cop” is? Corp.? Would seem best just to have “Zondervan,”
602 TheGospel in a Pluralist Society
Missing a space
The Gospel in a Pluralist Society
602-603 Grand Rapids, Michigan William B. Eerdmans Publ. Company Geneva Wcc 602 Publ.
Elsewhere, like line 610, “Michigan” does not appear after “Grand Rapids.” In any case, a colon should come after the place of publication. And probably “Publ. Company Geneva Wcc 602” is extraneous or at least unclear. Furthermore, the article submission cites an Eerdmans publication six times (in the footnotes and bibliography combined), with four different formats. It would seem the simplest would the format used with the Morris entry (and the Koester entry, minus the period after Grand Rapids.:)—that is:
Grand Rapids: Eerdmans.
623 Christianity Today
Periodical titles should be italicized
Christianity Today
Round 2
Reviewer 2 Report
I look forward to citing this in some work I am doing on Johannine polity.
Author Response
Thank you again for your attentive reading of my manuscript. I am very grateful.